# Perception of urban subdivisions in pedestrian movement simulation

**Gabriele Filomena**[1]*, **Ed Manley**[2,3], **Judith A. Verstegen**[1]

**1** Institute for Geoinformatics, University of Münster, Münster, Germany, **2** School of Geography, University of Leeds, Leeds, United Kingdom, **3** Leeds Institute for Data Analytics (LIDA), University of Leeds, Leeds, United Kingdom

* gabriele.filomena@uni-muenster.de

**Data Availability Statement:** The repository containing the Agent-based model presented in the paper and the necessary input data are available on Zenodo, http://doi.org/10.5281/zenodo.4318971. The workflow documenting the analysis of the

## Abstract

The perception of urban subdivisions, deriving from regionalisation processes and the identification of separating elements (barriers), has proven to dynamically shape peoples' cognitive representations of space and route choice behaviour in cities. However, existing Agent-Based Models (ABMs) for pedestrian simulation have not accounted for these particular cognitive mapping processes. The aim of this paper is to explore the behaviour of pedestrian agents endowed with knowledge about urban subdivisions. Drawing from literature in spatial cognition, we adapted a region-based route choice model, which contemplates a high- and a local planning level, and advanced a barrier-based route choice model, wherein the influence of separating elements is manipulated. Finally, we combined these two approaches in a region-barrier based model. The patterns emerging from the movement of agents employing such approaches were examined in the city centres of London and Paris. The introduction of regions in the routing mechanisms reduced the unbalanced concentration of agents across the street network brought up by the widely employed least cumulative angular change model (-.08 Gini coefficient). The inclusion of barriers further raised the dispersal of the agents through secondary roads, while leading agents to walk along waterfronts and across parks; it also yielded a more regular usage of pedestrian roads. Moreover, the region- and the region-barrier based routes showed deviation ratio values from the road distance shortest path (region-based: 1.18 London, 1.16 Paris, region-barrier based: 1.43 London, 1.33 Paris) consistent with empirical observations from pedestrian behaviour research. A further evaluation of the model with macro-level observational data may enhance the understanding of pedestrian dynamics and help tuning the interplay amongst urban salient elements at the agent level. Yet, we consider the movement flows arising from our current implementation insightful for assessing the distribution of pedestrians and testing possible interventions for the design of legible and walkable spaces.

## 1 Introduction

Urban form plays a crucial role in the emergence of flow of pedestrians [1, 2]. The configuration of the street network, the distribution of landmarks [3, 4], block permeability [5], and

ABM's results is also entirely available on Zenodo, http://doi.org/10.5281/zenodo.4318920.

**Funding:** We acknowledge support by Open Access Publication Fund of University of Münster. The funders had no role in study design, data collection and analysis, decision to publish, or preparation of the manuscript.

**Competing interests:** The authors have declared that no competing interests exist.

functional connotations [6, 7], shape what people find attractive [8, 9], pleasant [10–12], and spatially accessible [1]. Moreover, urban form entails a complex interaction between urban elements and creation of meaning [13]: on the one hand, the interaction between urban form and activity patterns generates interdependent [14] and hierarchical mental representations of space [15]; on the other hand, the human-environment interaction is mediated by preexisting representational schema of the environment that, in turn, reshape the urban form [16].

In *The Image of the City*, Kevin Lynch [3] introduces the notion of *imageability*, moving disciplines interested in the mind and disciplines interested in the geographical space close to each other. Imageability is a quality of the environment that relates to the ability of an observer to build a vivid and well defined mental representation of the environment—the *image of the city*. Lynch explores the relationship between urban form, mental images and how people navigate through a city [17]. Meaningful urban elements—paths, nodes, districts, landmarks and edges—are described as fundamental bricks that shape the representation of the city and influence individuals' spatial behaviour. The Lynchian elements have proven to have an effect on wayfinding and route choice processes [4, 18–22]: they allow giving sense, codifying and simplifying the external environment by means of categorisation and hierarchisation processes and, thereby, successfully navigating across the external environment.

Although the term urban form is used to indicate a large variety of fixed geographical elements that contribute to the spatial configuration of the city [23], research in pedestrian dynamics has mostly focused on the physical characteristics of the street network [24]. Measures of connectivity or centrality computed on the basis of path continuity [25, 26] and metric distance, in graph or axial representation of the street network, have turned out to be good predictors of movement flows of road users [1, 27–29].

This research direction has inevitably influenced the design of Agent-Based Models (ABMs)—a computational paradigm that allows analysing how variation in the mechanisms behind individual behaviour may generate different global patterns [30, 31]—for pedestrian movement simulation. Besides few cases [32, 33], landmarks', regions' and barriers' functions have not been incorporated in ABM approaches, despite the great opportunity offered by these models to portray genuine interactions between pedestrian agents and the urban environment [34]. Emphasis on street cost-related properties, as the main determinant of agents' behaviour [35–37], has led pedestrian ABMs to neither fully account for the complex interplay between the physical and mental layers in route choice processes [38], nor for the hierarchical organisation of the environmental knowledge [39, 40].

In this context, districts and edges—herein referred to as *regions* and *barriers*—shape the organisation of people's spatial knowledge; information about the external environment and its elements is stored in hierarchically organised sub-cognitive maps that are recalled at different stages of the route planning process [21, 22, 41]. The aim of this paper is to explore the behaviour of pedestrian agents endowed with knowledge about urban subdivisions, derived from the perception of regions and presence of natural and artificial barriers, using an ABM. Four ABM scenarios are designed to analyse the patterns emerging from the movement of agents making use of different route choice models: a) a currently employed approach based on road costs, i.e. least cumulative angular change, serving as the reference scenario, b) a region-based approach, c) a barrier-based approach, d) an integrated region-barrier approach. In the latter case, agents hold knowledge about the location of meaningful urban elements and their hierarchical relationship, and they formulate multi-level route plans by shifting from a coarse to a more grained representation of the city. The movement flows emerging from the agents' behaviour are compared across the four scenarios to answer the following research questions: a) How plausible are the routes formulated by means of the different route choice models? b) How does the incorporation of regions and barriers affect the dispersion of

pedestrians over the city? London and Paris are used as case study areas to examine the outcomes of the four scenarios models in different, yet comparable, urban contexts.

## 2 Background

In this section we elaborate on why we deem that the integration of regions and barriers in ABM may enhance pedestrian simulation models. In the first sub-section we present research regarding the role of regions and barriers in cognitive mapping and in shaping spatial knowledge. Thereafter, route choice processes involving the representation of regions are discussed, along with insights from urban design as regards the effect of barriers. Finally, existing ABMs of pedestrian movement in urban environments are discussed as pertains the inclusion of cognitive mapping processes.

### 2.1 The role of regions and barriers in cognitive mapping

Across the '70s and '80s, several studies advanced the idea that the knowledge about the external environment is organised hierarchically. This set of theories describes spatial knowledge as structured in 'nested level of details' [42]. People simplify and decompose the environment in essential building blocks by encoding spatial information under relations of containment (e.g. the Centre Pompidou is in 4th arrondissement or Tate Modern is located in Southwark) and ranking (e.g. a visually or culturally salient cathedral *vs* a non-salient terraced house). Such mechanisms, in turn, lead to a series of distortions in the perception and the representation of space, at different scales [41–45].

In the urban environment such organisation is the result of regionalisation processes [46] and the identification of barriers, and emerges throughout wayfinding and exploration activities [47]. Regions (districts) 'are the relatively large city areas which the observer can mentally go inside of, and which have some common character. They can be recognized internally, and occasionally can be used as external reference. (..) The characteristics that determine districts are thematic continuities which may consist of an endless variety of components: texture, space, form, detail, symbol, building type, use, activity, inhabitants, degree of maintenance, topography' [3]. The subdivision of an environment in regions derives from categorisation processes that take place in the general cognitive domain, for example when categorising social entities or objects [48, 49]. By considering certain attributes, either functional, thematic or geographical, environmental entities are grouped based on common properties or distinguished when dissimilar [46].

Conversely, barriers (edges) 'are linear elements not considered as paths: they are usually boundaries between two kinds of areas. They act as lateral references. Those edges seem strongest which are not only visually prominent, but also continuous in form and impenetrable to cross movement' [3, p. 62]. After Lynch's work, urban planners have further highlighted the role of barriers in structuring the image of the city and their contribution in the legibility of the urban form [50–52].

The idea that the cognitive representation of the city is structured hierarchically and resulting from collages [15] of several local cognitive maps meshed together, has been further held up by evidences from neuroscience and neuropsychology. The *cognitive-map hypothesis* [53] in the hippocampus has been extended to accommodate new components. *Boundary cells* [54, 55] support the functioning of the grid cells [56]. They codify distances from environmental barriers and support the comprehension of the environmental structure and its limits [57]. Furthermore, physical barriers and transition between regions indicate places where there is a change of perspective. On the one hand, such critical points lead to a fragmentation of the whole grid map in local maps [58, 59]; on the other hand, they elicit a shift from a cognitive

map to another, while the animal is navigating through the environment (*hippocampal remapping*) [60, 61].

## 2.2 The role of regions and barriers in route choice behaviour and pedestrian movement

The way people organise spatial knowledge has direct consequences on route choice behaviour processes. Chown and colleagues [21] claim that, for moving in complex and large environments, different space representations—*regional survey maps*—are employed and activated at transition nodes, *gateways*. These nodes are allocated along boundaries between regions: 'A gateway occurs where there is at least a partial visual separation between two neighboring areas. (..) They represent locations at which a choice will have to be made' [21]. A gateway is a 'vantage point' wherein new visual information allows the explorer to recall more defined regional survey maps. Other computational approaches to spatial knowledge in AI and robotics, such as TOUR [62] and Traveller [63] conceptualise similar regional abstractions in route planning.

Further evidence for the influence of regional subdivisions in route choice behaviour has been provided in recent studies by Wiener and coworkers [22, 64]. The authors observed that subjects, asked to complete navigational tasks in a virtual reality environment, would extensively make use of predefined regions to reach their different goals. The authors describe a *fine-to-coarse* heuristic that entails a continuous shift from detailed information, *place-connectivity*, to coarse space information, *region-connectivity*, when defining routes: regions constitute the higher level of a nested representation of space and they are employed to come up with a coarse plan; places belong to regions and they are conceptualised as decision points where local spatial representations are taken on and used for moving within a region.

The impact of artificial barriers, such as railways infrastructures or major roads, has been mostly discussed in transport planning with respect to pedestrian mobility, accessibility and interchanges between neighbourhoods: highways or roads with multiple lanes and poor crossing facilities constitute barriers both from a physical and a psychological perspective [65–67]. On the contrary, natural separating elements, like water bodies' banks or parks' boundaries, are associated with positive feelings [68], health and well-being [69]. Not only waterfronts, parks and green areas promote walkability [70–72] and determine environmental preference [3, 12, 68, 73, 74]; they also increase the chance of people living in their proximity to engage in walking behaviour [75–78] and they attract flows of pedestrians [76, 79].

## 2.3 Agent-based modelling for pedestrian movement simulation: Gaps

The first ABMs for pedestrian movement simulation in urban environments were advanced by Jiang [35, 80] (*SimPed*) and Schelhorn et. al [81, 82] (*STREETS*). SimPed, in line with Space Syntax theorisation, was designed on the assumption that the road configuration is the main predictor of pedestrian movement. In the authors' view, 'higher cognitive abilities are not required in the formation of movement patterns at a collective level' [35, p. 61]. Likewise, in STREETS, apart from a set of intermediate points along the shortest path between an origin and a destination, agents are not equipped with cognitive maps or symbolic representations of space. Even though in another model derived by Space Syntax theories agents' were equipped with information about the visibility of certain junctions [83], recent models [36, 84] mainly employ route choice approaches still devised on cost minimisation grounds—i.e. road distance, number of turns and least cumulative angular change. While Omer and Kaplan [36] manipulated the attractiveness of certain places on the basis of their land-use categorisation, thus stressing the relationship between configuration, movement and socio-economical

aspects, Esposito and colleagues [84] recommend integrating Space Syntax-based ABM with route choice behavioural theories derived from spatial cognition research.

Such an integration has been advanced in [85], where the effect of global and local landmarks in pedestrian movement is explored through an ABM. However, the effect of regions in route choice behaviour in ABM has been modelled only for simulating the behaviour of taxi drivers in London. Building on Wiener and Mallot's framework [22], Manley and colleagues [32] advanced a route choice mechanism that embodies different planning levels by representing an initial rough global plan, subsequently refined at higher granularity levels. However, no simulation model of pedestrians movement or traffic flows has accounted for the role played by barriers in shaping route formulation processes.

## 3 Methodology

Drawing from the works presented in the previous section, we designed an ABM for pedestrian movement simulation that investigates the effect of the cognitive processes, deriving form the perception of regions and barriers, in pedestrian navigation. We devised different scenarios so as to compare global movement patterns emerging from the individual routes of agents employing different route choice models. The role of regions and barriers was investigated, both by considering them as stand-alone components, and by combining their effect, in contrast to a purely cost-based scenario.

### 3.1 ABM and scenarios

In our ABM, agents represent pedestrians and complete a certain number of trips between pairs of origin and destination nodes (OD). Both origins and destinations were chosen randomly from the entire set of junctions, provided that they were as distant in meters (Euclidean distance) as a random number between 1000 and 3000. These distances were intended to generate routes whose length is higher than average walking trip's length (for example, Yang and Diez-Roux [86] report an average length of 1250 m. for walking trips in the US population) and that require pedestrians to employ the urban elements here modelled; such routes would represent itineraries walked by tourists, urban explorers, people engaged in different activities throughout the day or people reaching novel destinations. The model was run for four scenarios in which different route choice models were used to regulate the agents' behaviour (Fig 1):

1. Least cumulative angular change (city-level). As this is a purely cost-based scenario, widely employed in existing modelling approaches, we use it as a reference scenario, *AC scenario*;

2. A region-based route choice model, combined with a local heuristic (least cumulative angular change, intra-region), *RB scenario*.

3. A barrier-based route choice model, combined with least cumulative angular change (city-level), *BB scenario*.

4. A region-barrier based route choice model, combined with a local heuristic (least cumulative angular change, intra-region), *RBB scenario*.

The model consisted of 2000 agents formulating one trip each across a case study area. Each scenario was executed five times to average out the randomness introduced by the random selection of OD pairs and the stochastic functions in the route choice models (explained hereafter). The number of runs was fixed at five as we observed little variation in the emerging patterns after this number of executions.

**AC scenario: Least cumulative angular change scenario.** The AC scenario aims to serve as the reference scenario by incorporating a route choice model widely used in existing ABMs

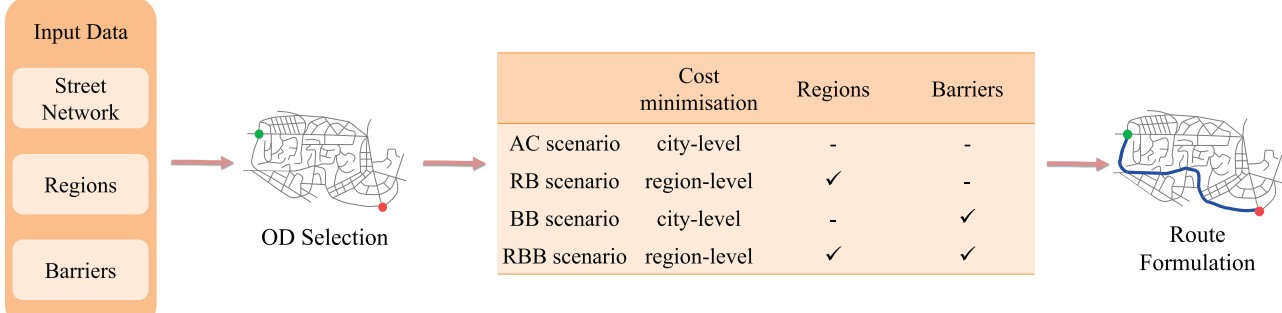

**Fig 1. The conceptual model of the ABM.** The street network (nodes and links), regions and barriers are loaded in the simulation. A set of OD pairs is defined per each run and assigned to the scenarios where agents complete trips on the basis of different route choice models. The AC scenario is based on cumulative angular change minimisation; the RB scenario incorporates a region-based route choice model; the BB scenario, a barrier-based route choice model; in the RBB scenario, agents make use of a combination of the three approaches, combining the effect of regionalisation processes, barriers and least cumulative angular change.

for pedestrian simulation. Agents seek to take the route with the least cumulative angular change [28] between the origin and the destination. The Dijkstra's shortest path algorithm [87] is employed to formulate the routes in a dual graph representation of the street network wherein links—representing street intersections—are weighted with the angle of deflection between the corresponding road segments, and nodes represent street segments. Angle costs are modelled as stochastic variables in order to incorporate people's cognitive inaccuracy in perceiving and remembering turning angles [88, 89]. The perceived cost $i_e$ of a street segment $e$ is defined as:

$$i_e = Z_f$$
$$with\ Z_f \sim N(cost_e, (0.10 * cost_e)^2)$$

(1)

where $cost_e$ is the actual cost of the segment, and $Z_f$ is a standard normal distribution. Hence, the deviation from the actual cost is relative, i.e. the perception error is typically larger for more abrupt turns. This approach introduces a probabilistic component in agents' the route choice behaviour.

**RB scenario: Region-based route choice model.** Regions were extracted from the entire street network of the two case study areas, by employing the modularity optimisation algorithm [90]. The algorithm optimises modularity [91], a measure of global strength of a certain division in partitions. From the street morphology, such an algorithm identifies regionalised structures that overlap with peoples' perceived urban subdivisions [92] and whose homogeneity is grounded on socio-geographical characteristics [93]. In our work, the membership of the junctions to certain network partitions (regions) was inferred from divisions deriving from topological ties in a dual graph representation.

In this scenario, agents make use of the so-extracted regions and employ a route choice model presented by Manley and colleagues [32] and adapted for pedestrian movement [33]. Agents subdivide the urban environment in portions cognitively easy to handle, and employ them to organise knowledge about other elements (i.e. streets, junctions). These elements are allocated to *sub-regional maps*, activated at certain *gateways* ([21], see section 2.2). The region-based route choice model is structured in two planning levels (following [22]):

1. *Coarse plan—sequence of regions*: a rough sequence of regions is first formulated on the basis of the gateways' location and the agent's direction of movement. This is a high-level plan that takes place before navigating the environment

2. *Fine plan—intra-region path*: when transitioning to a new region, the agent activates the corresponding sub-regional map and retrieves information about possible intermediate junctions and street segments, so as to define its intra-region route.

Specifically, when the agent is about to start a new route, it firsts formulates a route (coarse plan) that only takes into account the position of junctions along borders between regions (gateways). A gateway consists of a pair of nodes: an *exit node* from a certain region and a linked junction that belongs to an adjacent region, an *entry node*. From its *current location* (except for the first iteration, this term refers to the future location of the agent in that region; at this stage, this is a mere cognitive plan and the agent is not actually located there), the agent selects the next region to visit, by choosing the most favourable gateway amongst viable candidates. A gateway pair is considered viable when it meets the following conditions (Fig 2):

- The distance between the possible exit node and the destination node is shorter than the distance between the current location and the destination.

- The exit and entry nodes are in the direction of the destination; the angle ($\alpha$) formed by the current location and a candidate exit node is supposed to be $\alpha >$ angle(*currentLocation*, *destinationNode*) − 70° and $\alpha <$ angle(*currentLocation*, *destinationNode*) + 70°. Although Manley [32] employed a ±90° constrain, we used a narrower space, determined subjectively. We claim that pedestrians may not be as willing as car drivers to take long detours. However, when no gateways can be found within a 140° cone towards the destination, the agent considers gateways within a 180° cone instead.

- The entry node belonging to the next possible region is in the direction of the destination as well.

The agent selects the least deviating gateway pair amongst the ones that satisfy the conditions; it repeats such a process from the new entry node, so to move forward in the formulation of the regions' sequence, till the region of the destination node is reached.

a) Gateways assessment

b) Picking the most promising gateway and moving on to the next region

c) Finalising the coarse plan (sequence of regions and gateways)

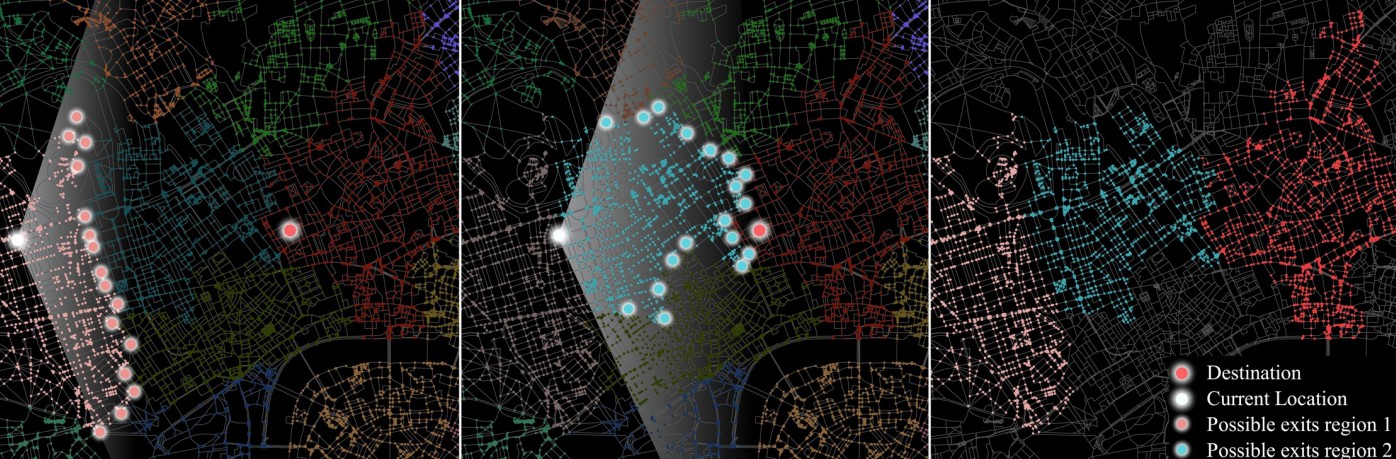

**Fig 2.** The gateways selection and formulation of a coarse plan (for visualisation purposes, not all the viable exit nodes are shown): a), b) The agent cognitively moves from a region to another, till c) the destination's region is reached, on the basis of the gateways' locations. Data source (street network): OpenStreetMap data [94].

Throughout the actual navigation, the fine plan takes place when the agents enters a new region, i.e. arrives at an entry node or starts from the origin, after the coarse-plan. Sub-regional maps are recalled and used to retrieve information about the street segments' costs and relationship amongst different locations; the agent makes use of a local minimisation heuristic (LMH)—least cumulative angular change within the region—to fill the gap between the current entry node and the next exit node. The agent navigates across the region, reaches the gateway and calls the next sub-regional map.

**BB scenario: Barrier-based route choice model.** To identify barriers, the following features were extracted within the case study areas:

- Water bodies: rivers, lakes (larger than a certain threshold defined by the modeller), canals or sea coasts (N/A).

- Parks: parks at least large 10000 $m^2$.

- Railways: over ground tracks, including super-elevated structures or bridges.

- Major roads: roads categorised as 'trunk', 'motorway', or 'primary' in OpenStreetMap.

We claim that barriers not only influence regionalisation processes and shape the overall structure of the mental image [50, 40], but they can also play a role in route choice decisions. While water bodies and parks exhort walkers to wander along them, major roads and railways structures may keep pedestrians away or discourage people to walk along them for too long. We here define the first as *natural barriers* and the latter as *severing barriers*, on the basis of research around barriers and walkability (see section 2.2). Along these lines, in this scenario the agent alternate the identification of salient orienting barriers with navigation between pairs of nodes. More precisely, from a current location (initially the origin):

- The agent looks for barriers in a 70° research cone towards the destination (this is a narrower research space than the one defined for the gateways; it is meant to prevent extreme detours as no directional ranking is performed on barriers, see below). Barriers that are more distant from the current location than the destination are disregarded.

- When more than one barrier is identified, the farthest water body barrier is chosen; if no water bodies are identified, the agent picks the farthest park barrier, if any, or, otherwise, the farthest viable severing barrier.

- The closest node to the agent's location, amongst the ones adjacent to the identified barrier, is selected (*barrier sub-goal*) and reached by minimising cumulative angular change at the global level, i.e. without considering regional subdivisions.

- The agent reiterates the previous steps from the new location. If no more barriers are detected, the agent finally moves to the destination, again, by minimising cumulative angular change.

Furthermore, while natural barriers reduce the perceived cost of the segments along or within (in the case of parks), severing barriers increase the perceived cost of the segments aside them (or of the segments themselves), discouraging pedestrians to walk along them. In such cases, the perceived cost $i_e$ of a street segment $e$ is defined as:

$$i_e = cost_e * Z_d$$

$$with \ Z_d \sim \begin{cases} min(N(0.70, \ \sigma^2), 1.00), & \text{if } e \text{ lies along or within a natural barrier} \\ max(N(1.30, \ \sigma^2), 1.00), & \text{if } e \text{ constitutes or lies along a severing barrier} \end{cases} \quad (2)$$

where $cost_e$ is the actual cost of the segment; $Z_d$ is a distribution with a mean of 0.70, when the

segment lies along natural barriers, or 1.30, when it lies along severing barriers, and a variance ($\sigma^2$) of 0.01. We subjectively chose the mean values (0.70 and 1.30) as they proved to elicit differences in the agents' behaviour without affecting the plausibility of the routes. The *min()* and *max()* are applied to avoid that by chance severing barriers are positively perceived and vice versa. For all the other segments, not adjacent to any barrier, costs are modelled as in Eq 1.

## 3.2 RBB scenario: Combining the effect of regions and barriers in route choice behaviour

Finally, we advance a route choice model based on both types of urban subdivisions, that builds upon the region- and barrier-based navigation approaches, thus combining the effect or regionalisation processes and perception of urban barriers in spatial behaviour. Such a route choice model is structured in three interacting planning levels that move from an initial coarse plan to local, finer, decisions.

In this scenario, agents are equipped with a symbolic representation of the urban environment—the *city cognitive map*—that includes information about street segments and junctions (AC scenario), regional subdivisions (RB scenario) and knowledge about the position of barriers (BB scenario). This approach makes use of the already introduced notions of gateways and sub-regional maps, and extends the RB scenario in light of evidences regarding the role of barriers in inducing the retrieval of sub-maps or new frame of references (see section 2.1). The region-barrier based route choice model is structured as follows:

1. *Coarse plan: sequence of regions—from RB scenario*: a rough sequence of regions is formulated (Fig 2). In this scenario, this high-level plan can be partially revised throughout the next stages.

2. *Fine plan: barriers identification within the region—from BB scenario*: this stage takes place when the agent enters or starts in a new region; the agent activates the corresponding sub-regional map and retrieves information about main nodes and barriers (Fig 3a). A barrier sub-goal is possibly identified and the coarse plan is fed with new information (Fig 3b).

3. *Fine plan: intra-region path formulation—from RB scenario*: the agent adopts a LMH (i.e. least cumulative angular change) whereby road costs are taken into account to define an intra-region path, either between the current location and the exit node or between the current location and a barrier sub-goal (Fig 3c). In the latter case a further readjustment takes place.

4. *Fine plan: potential readjustment*: When reaching barrier sub-goals potentially identified in the previous stage, a more favourable exit node could be picked (Fig 3d). The *Fine plan: intra-region path formulation* occurs again, between the sub-goal barrier and the exit node, prior to navigation.

In addition, when making use of the LMH, the costs of the street segments are perceived by the agent as in the BB scenario (Eq 1).

## 3.3 Scenario evaluation

Although the study of complex social systems is still characterised by a large debate around what constitutes a standardised approach to ABM validation [95], it is agreed upon that various emergent patterns at the macro-level should be evaluated to assess the model outcomes [96]. The deviation of pedestrian routes from the road distance shortest path has raised interest in route choice behaviour research, both to understand to what extent actual routes are longer than the shortest path and to identify the factors that may explain such divergence [97, 98].

a) Recalling sub-regional map and original exit gateway  b) Veryfing the presence of orientining barriers  c) Identification of the sub-goal and exit reassessment  d) Intra-region route formulation and transition

**Fig 3. Fine plan: Intra-region route with barriers identification.** a) At the region's entry, the agent recalls the congruent sub-regional map and b) identifies possible barriers that could further help the navigation across the region; c) the agents moves towards the corresponding barrier sub-goal and reassess the exit nodes, on the basis of the new position and orientation towards the destination; d) the agent reaches the chosen exit-node and activates the new sub-regional map. Data source (street network): OpenStreetMap data [94].

Although, as discussed above, we deem route choice models purely based on cost minimisation partial and not able to account for the complexity and variability of human spatial behaviour, people do make use of metric information [99, 100] and, successfully or not, may attempt to minimise distance or change of direction [101]. Hence, to assess the plausibility of the routes generated within the four scenarios, we computed for each route the deviation from the corresponding *road distance shortest path* (SP) as:

$$d_r = \frac{length_r}{length_s} \tag{3}$$

where *r* is a route in a certain scenario, between an OD pair, and *s* is the shortest route, computed on the basis of road distance in a primal graph representation of the street network (employing the Dijkstra's shortest path algorithm [87]), between the same OD pair. For each scenario a median deviation ratio is obtained.

Furthermore, the outcomes of the model were assessed by comparing the patterns resulting from the four different scenarios with respect to the distribution of the agents across the street network, at the city-level (pedestrian traffic volume). Such evaluation was accompanied by a measure of dispersion, the Gini coefficient of inequality. The Gini coefficient is a global measure of inequality ranging between 0 (perfect equality, widely dispersed) and 1 (perfect inequality, very concentrated) [102, 103]. It is computed as a relative mean difference of all pairs of observations:

$$G = \frac{\sum_{i=1}^{n} \sum_{j=1}^{n} |x_i - x_j|}{2n^2 \bar{x}} \tag{4}$$

where $x_i$ and $x_j$ are the observed value for the variable $x$ at two different locations, $n$ is the number of observations and $\bar{x}$ is the mean value of the variable. Although, this coefficient was introduced for investigating socio-economical inequality across different geographical areas, it 'can be reinterpreted as measuring the amount of concentration of the values on the map' [104,

p. 60]. The coefficient was computed for each scenario, on the median number of agents per street segment over the five runs.

At the case study level, we assessed a set of variables to investigate whether the introduction of regions and barriers, as well as their combination, in the agents' individual behaviour would result in different global patterns:

- Portion of pedestrian streets in walked routes (street segments tagged as *footway*, *pedestrian*, *living street*, *path* in OpenStreetMap), at the city level.

- Portion of major roads in walked routes (street segments tagged as *primary roads* in Open-StreetMap) at the city level.

- Portion of streets along natural barriers in walked routes.

The Games-Howell post-hoc test [105], a non-parametric alternative for the *t-test* that can be employed to compare multiple groups with unequal variances, was used to assess whether differences in these variables between scenarios were significant (with a.01 *p-value*). As these variables are computed as percentages over the routes' length, the outcomes would not depend on the length of the routes.

### 3.4 Case study areas

The cities of Paris and London were chosen as case study areas. This is to observe how different city structures and their urban elements mould the behaviour of the agents in the different scenarios. Paris and London are both crossed by rivers, the Seine and the Thames respectively, that divide the urban structure in two macro-regions and likely shape the cognitive representation of the city of their inhabitants' and visitors'. Moreover, the development of a dense railway structure and the presence of monumental railway stations have broken movement continuity in these two cities and brought up 'awkward confrontations' with the street network [106]. Paris and London are amongst the most highly populated urban areas in Europe and they present a mix of old and modern architecture. Although similarly inhabited, they are characterised by 'divergent spatial organisations' [107], especially in relation to population density. Paris aligns with a European idea of urban structure, where people's activities converge in the centre. London, conversely, exhibits a lower concentration of people with a more spread distribution of residents and jobs across its extent [107].

A bounding box (8000 x 8000 m, Fig 4) was used to extract a central region from the Greater London Area (the administrative boundary of London, composed of 33 local government districts) and the Métropole du Grand Paris (this is the area that includes Paris and *la petite couronne*, i.e. the departments surrounding Paris: les Hauts-de-Seine, la Seine-Saint-Denis and le Val-de-Marne). The regions modelled in the ABM were extracted from the street network (OpenStreetMap data [94]) of the Greater London Area and the Métropole du Grand Paris (see section 3.1, RB scenario).

## 4 Results and discussion

### 4.1 Routes' plausibility evaluation

Cumulative angular change minimisation (AC scenario) generated the shortest trips across the route choice models incorporated in the scenarios, both in London and in Paris (1.13 and 1.12 times longer than the SP, Table 1). Predictably, regions and barriers caused, to different extents, detours and longer paths. The routes in the RB scenario (for regional subdivisions see S1 Fig, in the S1 Appendix) are slightly longer than the ones in the AC scenario (around 4% in London and 3% in Paris), 1.18 and 1.16 longer than the corresponding SP. The definition of a

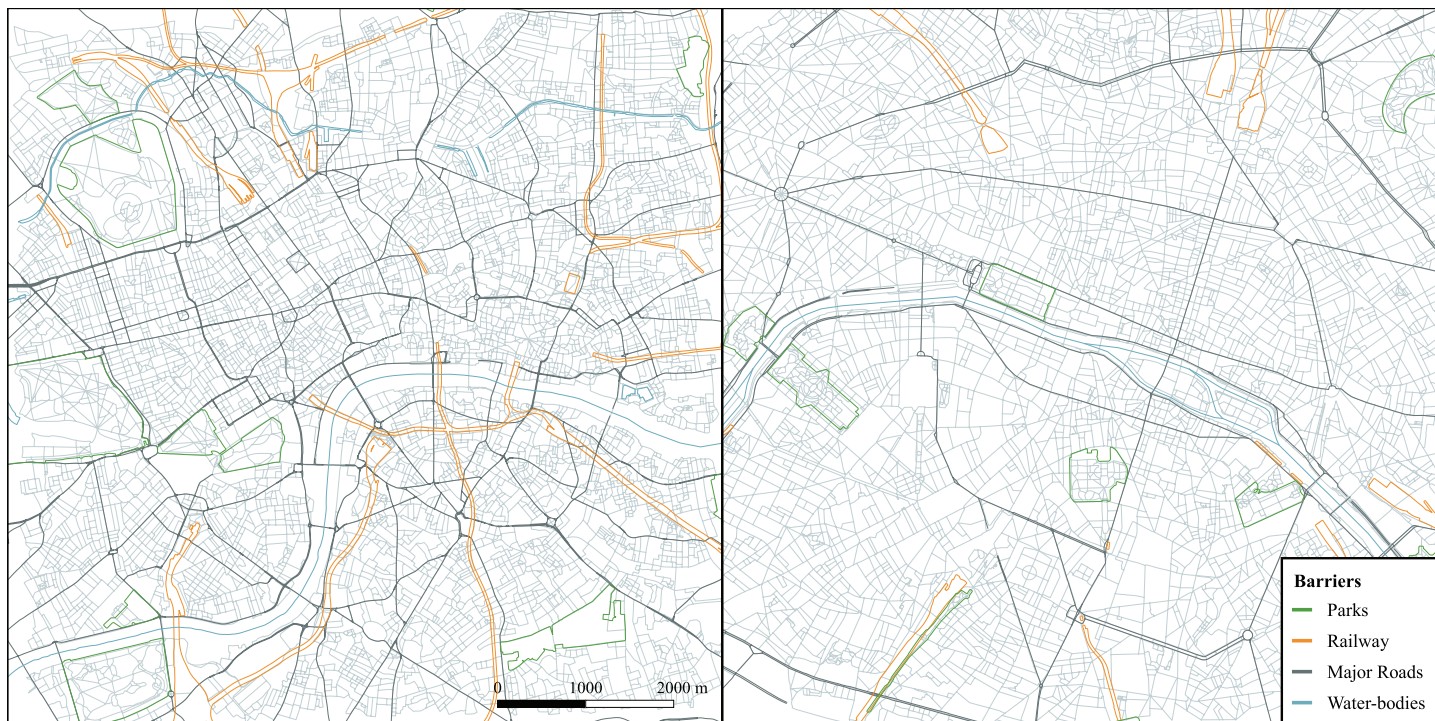

**Fig 4. The case study areas: London (UK), left, and Paris (FR), right, bounding box 8000 x 8000 m; walkable street network and barriers.** The map is oriented north. Data source (street network, railways, parks and water bodies): OpenStreetMap data [94].

sequence of regions brought agents to take alternative roads and partially deviate from the continuity of the routes generated when minimising cumulative angular change globally. In this sense, a LMH may conduce to local optimisation (e.g. minimising angular change between an entry and an exit node of a certain region), while causing an increase in angular change at the global level; this applies to the RBB scenario as well. Purely barrier-based route choice agents,

**Table 1. Statistics of the ABM scenarios' routes.**

| | London | | | |
| --- | --- | --- | --- | --- |
| | AC | RB | BB | RBB |
| Median route length | 3033.30 m. | 3166.84 m. | 4312.93. | 3766.10 m. |
| Median $d_r$ from SP | 1.13 | 1.18 | 1.67 | 1.43 |
| % agents using the SP ($d_r <= 1.10$) | 40% | 27% | 4% | 7% |
| $\rho$ route length *vs* deviation | .48 | .61 | .73 | .67 |
| | Paris | | | |
| | AC | RB | BB | RBB |
| Median route length | 2899.78 m. | 2988.50 m. | 3718.26 m. | 3425.98 m. |
| Median $d_r$ from SP | 1.12 | 1.16 | 1.47 | 1.33 |
| % agents using the SP ($d_r <= 1.10$) | 41% | 34% | 10% | 14% |
| $\rho$ route length *vs* deviation | .48 | .57 | .74 | .67 |

*AC*: Least cumulative angular change scenario; *RB*: Region-based scenario; *BB*: Barrier-based scenario; *RBB*: Region-barrier based scenario. *SP*: road distance shortest-path; $\rho$: Pearson product-moment correlation coefficient.

only partially constrained to move in the direction of the destination, incurred numerous deviations from the destination (routes 1.67 and 1.47 times longer than the corresponding SP). The RBB scenario inherited this property from the barrier-based route choice model. Yet, the employment of regions in this scenario coerced the search of barriers, thereby generating shorter routes than the BB scenario (1.43 and 1.33 times longer than the SP).

Overall, routes in Paris deviate less from the shortest path in the RB, BB and RBB scenarios. We believe this might be due to a different distribution of the barriers modelled in London and in Paris. In Paris, some barriers, because of their location, may be reinforcing each other's separating nature (e.g. railways and primary roads developing along the Seine; parks mainly located in the proximity of the river). Besides, the regional subdivision emerging from the street network (e.g. the regions' extension, the number of regions identified) may impact the length of the routes and their deviation from the SP.

The results of the RBB scenario seem promising and align with findings in literature on pedestrian route choice behaviour: Foltête and Piombini [97] observed a 1.33 deviation ratio ($d_r$) from the shortest path (shortest route 526 m, average route length 700 m), with deviations up to 1.74, across 257 pedestrian routes in Lille (France). Such deviation ratio is equal to the RBB scenario in Paris (1.33) and indicates a good plausibility of the routes depicted by the RBB agents. Considering evidences supporting the idea that longer routes are more likely to present deviations [98, 108, 109], one can assume that the deviation ratio reported in [97] would be even higher for the routes generated by our ABM (OD pair separated by distances ranging from 1000 to 3000 m). In view of this, the deviation ratio of the RBB scenario's routes in London (1.43) can be considered plausible. On the contrary, the AC scenario, and partly the RB scenario, results in a deviation ratio that we consider too low, especially bearing in mind the length of the modelled routes. The deviation ratios of such scenarios are however comparable with the one observed in another study conducted for very short trips (around 548 m long, deviation ratio 1.15) [110].

Moreover, although in [97] 55% of the trips did not involve any deviations, other works reported higher share of pedestrians relying on the shortest route. In an experiment conducted in West Jerusalem (Israel), around 73% of the sample employed the shortest path [111], whereas in two different areas in San Francisco (USA) around 68% of the pedestrians chose the shortest path towards transit stations [110]. In our model, percentages are much lower when using a 1.10 tolerance (as suggested in [112] for analysing drivers' route choice). Only 7% (London) and 14% (Paris) of the agents in the RBB scenario adhered to the shortest route (see Table 1). On the contrary, in the AC scenario 40% and 41% of the trips followed the shortest route, thus showing percentages closer to the findings mentioned above.

Fig 5 shows the variation of the deviation ratio from the SP, on the basis of the number of regions traversed along the routes. Little intra-scenario differences emerge in the deviation from the SP when comparing routes through 1 region and routes through 2 regions; only the RB and BB scenarios have a statistically significant difference in deviation ratio for routes that traversed 1 and 2 districts (London: RB +0.05, BB +0.09; Paris: RB +0.04, BB + 0.09). Routes through 3 regions present higher deviation ratio values than the routes, in the same scenarios, that crossed 1 or 2 regions (e.g. increment from 2 to 3 regions, London: AC +0.04, RB +0.04, BB +0.23, RBB +0.06; Paris: AC +0.03, RB +0.05, BB +0.14, RBB +0.05); likewise, routes through 4, 5 or more regions present statically significant larger deviations as compared to routes that crossed fewer regions (e.g. increment from 3 to 4 regions, London: AC +0.06, RB +0.07, BB +0.19, RBB +0.12; Paris: AC +0.06, RB +0.09, BB +0.14, RBB +0.14). Therefore, generally speaking, the number of regions traversed along the agents' routes increased the deviation ratio. To explain such pattern, one can conceive the number of regions as an index of route complexity, both in relation to the large distance separating an origin and a destination,

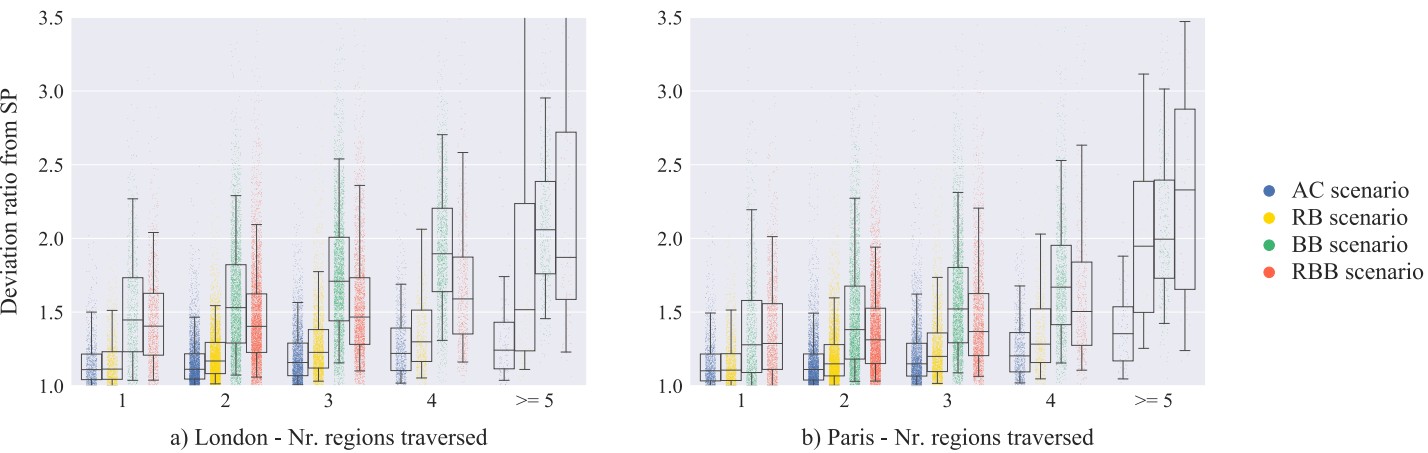

**Fig 5. Distribution of the routes' deviations ratio from the road distance shortest-path (SP), on the basis of the number of regions traversed along the routes.**

and the cognitive effort required to deal with different urban areas (i.e. regions featured by diverging street configuration, building densities, etc. [113]). Complexity and long sequences of sub-goals (e.g. gateways or barriers) may cause people to take longer detours, as a result of disorientation or inability to follow an optimal path [114]. Yet, to our knowledge, the relationship between the number of areas or regions traversed in an environment and the deviation from the shortest path has not been explicitly explored.

Finally, across all the scenarios, a strong correlation between the deviation ratio from the SP and the routes' length arose (Table 1, Fig 6). The BB and RBB scenarios, which present the largest average deviation ratio, also exhibit the highest correlation coefficients between the length of the routes and the deviation ratio: 0.73 (BB) and 0.64 (RBB) in London, 0.74 (BB) and 0.67 (RBB) in Paris. Longer routes are more prone to be featured by higher number of regions and to possibly encounter more barriers (darker markers' colours, Fig 6), thus defining more complex sequence of sub-goals, and, in turn, incurring in numerous deviations. In total, most of the routes are featured by deviation values lower than 1.5 (59% in London and 68% in Paris) and the majority of the routes with considerable deviation ratios (median > 1.5) are longer than 4000 m.

This is in line with research on route choice behaviour: the relationship between long routes and increases in the deviation from the SP has been reported both for pedestrians [98, 108, 109] and car drivers [112, 115]. Despite the fact that we did not find a strong correlation between the portion walked along and across natural barriers and the deviation from the SP, highly deviating routes seem to be featured by high portions in proximity of such urban elements (Fig 6, variation in markers' colours). Research on walking behaviour has stressed how preference for friendly pedestrian facilities and street layouts [108, 110], pleasant views and natural landscapes [97, 98, 108] may induce people to take longer routes, thus causing deviations from the SP.

### 4.2 Agents' distribution across the street network

The four route choice models implemented in the scenarios yielded different movement flows both in London and in Paris (Figs 7 and 8, Table 2). In London, the AC scenario depicts a very clear skeleton of lines of movement that overlaps with major roads across the city centre and, at times, its boundaries (London Inner Ring Road). In Paris, apart from some roads

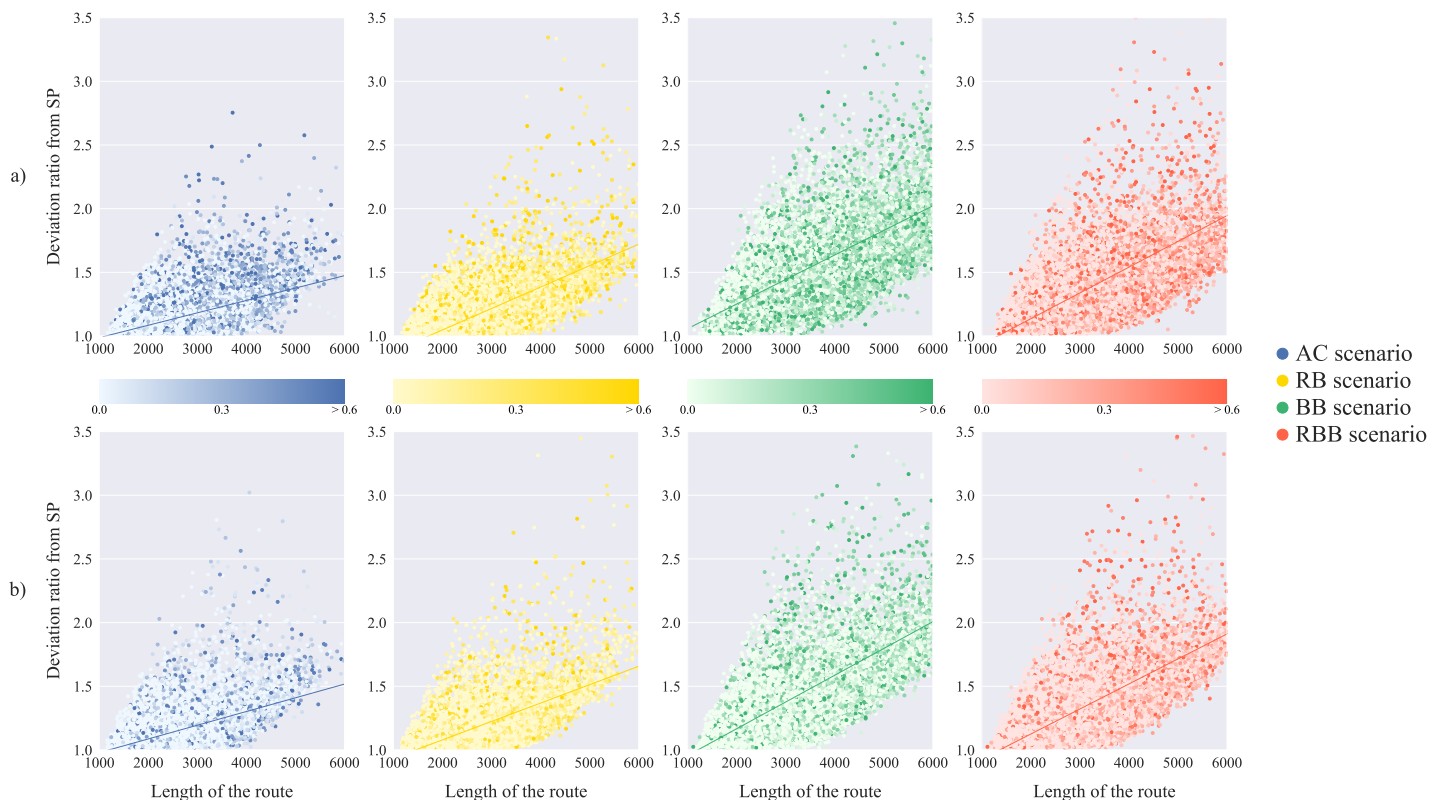

**Fig 6. Length of the ABM scenarios' routes *vs* deviation ratio from the road distance Shortest Path (SP).** *Saturation* and *value* in the markers' colours indicate the portion of the routes walked along natural barriers (see legend). The lines are regression lines between the two variables.

connecting the south with the north, the agents are distributed across a larger set of roads. The least cumulative angular change model generated the least dispersed distribution (highest Gini coefficient) of agents over the street network (London: 0.79; Paris: 0.71).

In London, while major roads emerge as dominant channels of movement—the scenario routes' present the highest portion walked along major roads across the ABM (median 26%, Fig 9) -, secondary roads are employed only marginally. Despite the movement of some agents along the south-bank of the river (up to 2.6% of the agents), the AC scenario shows in London a low portion of pedestrian roads (11%) and paths along rivers and parks (2%). On the one hand, major roads appear to induce gentle changes of direction; on the other hand, secondary roads across London, for their relationship with the rest of the network, do not facilitate movement on the basis of angular continuity. In Paris, albeit the AC scenario still generated the least dispersed distribution of agents, the range of roads traversed is wider, as compared to London. This may occur as the urban structure appears more complex, although not as dense as some areas in London. Nevertheless, also in Paris the percentage walked along pedestrian roads over the routes (7%) is significantly lower than all the other scenarios (Fig 9).

In London, the movement flows emerging from the RB scenario partly resemble the ones generated by the AC scenario. Volumes along primary roads in the central part and in the east are still rather high, whilst the west, the north and part of the south are characterised by a more even distribution. However, the routes of this scenario display significantly lower portions walked along major roads (18%), as compared to the AC scenario (-8%, Fig 9). In Paris,

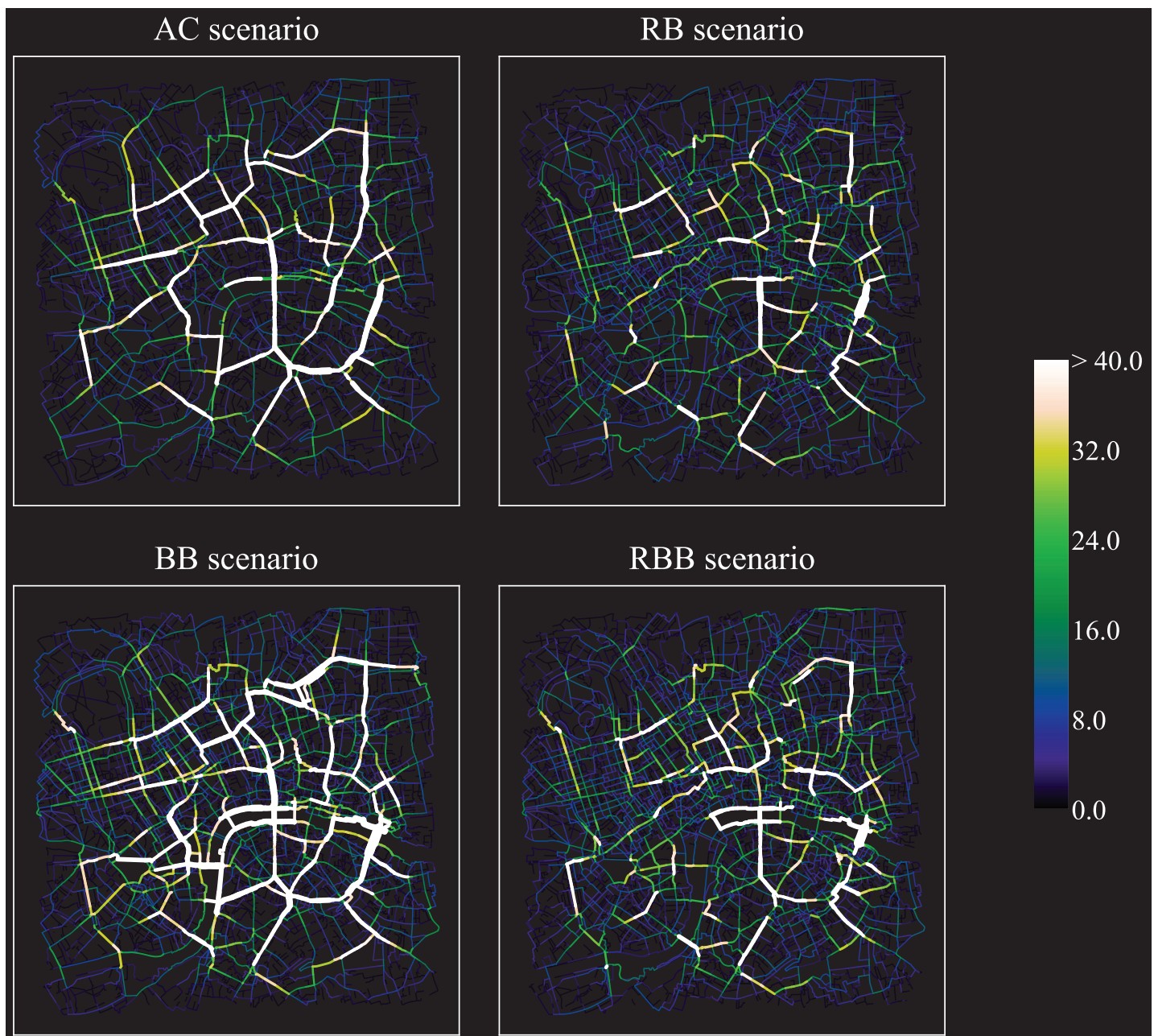

**Fig 7. Movement flows of agents across the street network resulting from the four ABM scenarios (nr. of agents per street segment, median across runs): London.**
*AC*: Least cumulative angular change scenario; *RB*: Region-based scenario; *BB*: Barrier-based scenario; *RBB*: Region-barrier based scenario. Data source (street network): OpenStreetMap data [94].

agents rarely converged in the same streets, but rather distributed across secondary roads, especially in the north and in the centre (in proximity to the river). The Gini coefficient demonstrates a substantially lower concentration of agents as compared to the AC scenario (London: -0.09; Paris: -0.07). In this sense, the introduction of regions may have mediated the low dispersal deriving from exclusively minimising angular change. Interestingly, apart from the north riverbank in London (traversed in the RB scenario by up to 2.6% of the agents, as in the

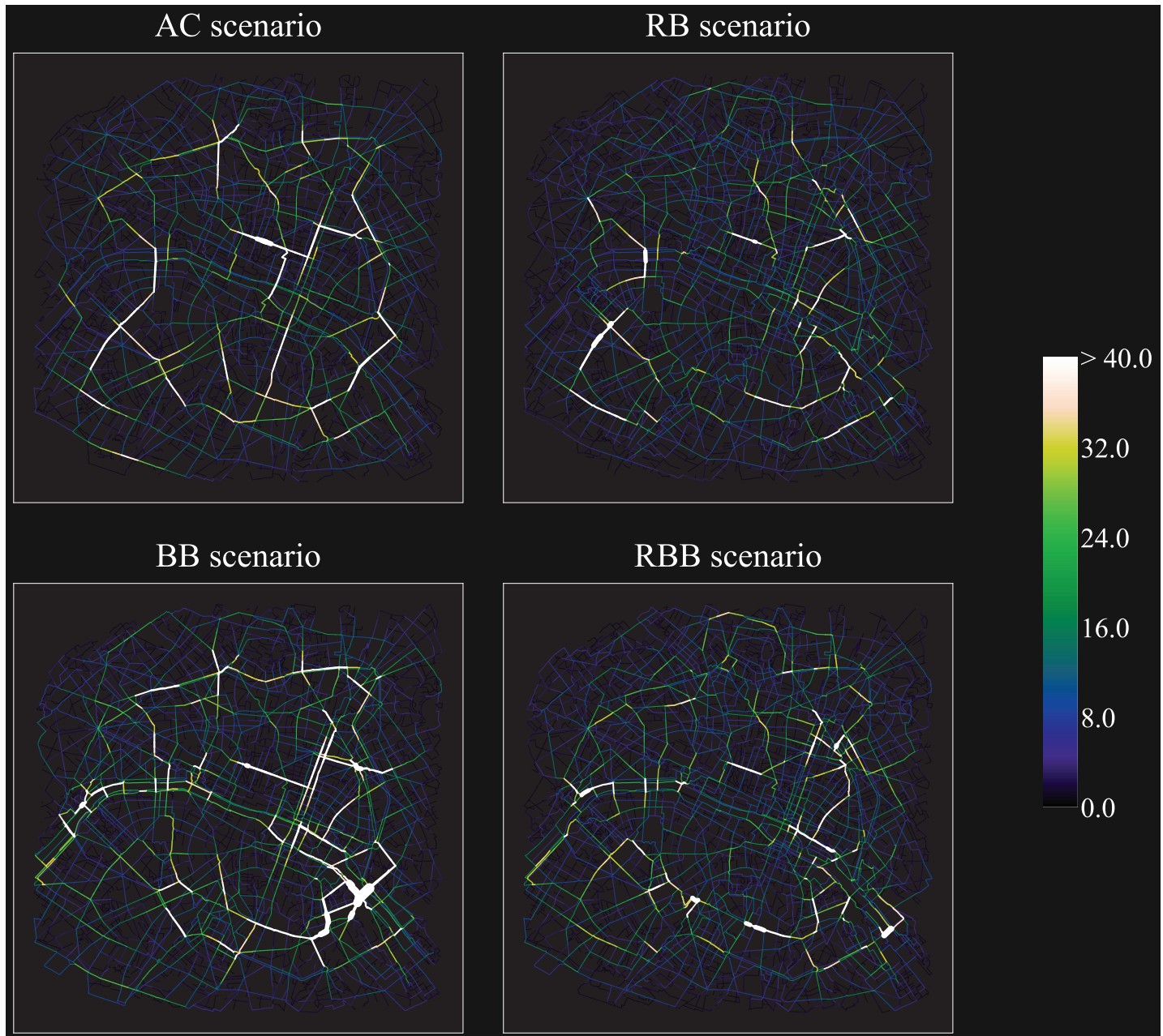

**Fig 8. Movement flows of agents across the street network resulting from the four ABM scenarios (nr. of agents per street segment, median across runs): Paris.**
*AC*: Least cumulative angular change scenario; *RB*: Region-based scenario; *BB*: Barrier-based scenario; *RBB*: Region-barrier based scenario. Data source (street network): OpenStreetMap data [94].

AC scenario), waterfronts are generally not walked along in the RB scenario, neither in London nor Paris. As a consequence of the regional subdivision, rivers become boundaries to cross and parks are only traversed if placed along angular advantageous paths.

Conversely, in the BB scenario, in London, the river was walked along by up to 4.7% of the agents. Similar volumes surface along a canal (north-east, up to 4.5% of the agents), park boundaries (north-west and west, 3.3%) and major roads (peaks of 8.1% in the east). Agents,

**Table 2. Statistics of the ABM scenarios.**

| | London | | | |
| --- | --- | --- | --- | --- |
| | AC | RB | BB | RBB |
| Mean nr. agents per street | 4.50 | 4.98 | 6.57 | 6.01 |
| Stdv. nr. agents per street | 9.94 | 8.21 | 11.71 | 9.13 |
| Max nr. agents per street | 98.00 | 124.00 | 131.00 | 128.0 |
| Gini coefficient | 0.79 | 0.70 | 0.72 | 0.67 |
| | Paris | | | |
| | AC | RB | BB | RBB |
| Mean nr. agents per street | 4.84 | 5.21 | 6.31 | 5.88 |
| Stdv. nr. agents per street | 8.28 | 7.42 | 9.83 | 8.02 |
| Max nr. agents per street | 70.00 | 66.00 | 130.00 | 77.00 |
| Gini coefficients | 0.71 | 0.64 | 0.68 | 0.63 |

*AC*: Least cumulative angular change scenario; *RB*: Region-based scenario; *BB*: Barrier-based scenario; *RBB*: Region-barrier based scenario.

although meant to overestimate costs of primary roads, made large use of them, in particular in the east, south and central part of the city. In Paris, the whole extent of the river is featured by flows of movement along its banks. In the east, several roads aside the river were crossed by around 2.5% of the agents, up to 4.5%; same in the west, where some streets were crossed by up to 3.6% of the agents. Yet, overall, the agents' distribution partly resembles the one emerging from the AC Scenario. In some cases, major roads present more substantial volumes, as demonstrated by the highest portion walked along this type of roads in Paris (13%, Fig 9): agents may have been pushed more often towards this type of barrier, due to the scarcity of other barriers in this case study area. As compared to London; parks are also mainly located along the river itself (see Fig 4) and are less ample.

Finally, the RBB scenario shows a pattern that builds partly on the RB scenario and partly on the BB scenario. As in the RB scenario, both in London and Paris, agents walked through secondary roads and a higher number of roads than the AC scenario. However, water bodies

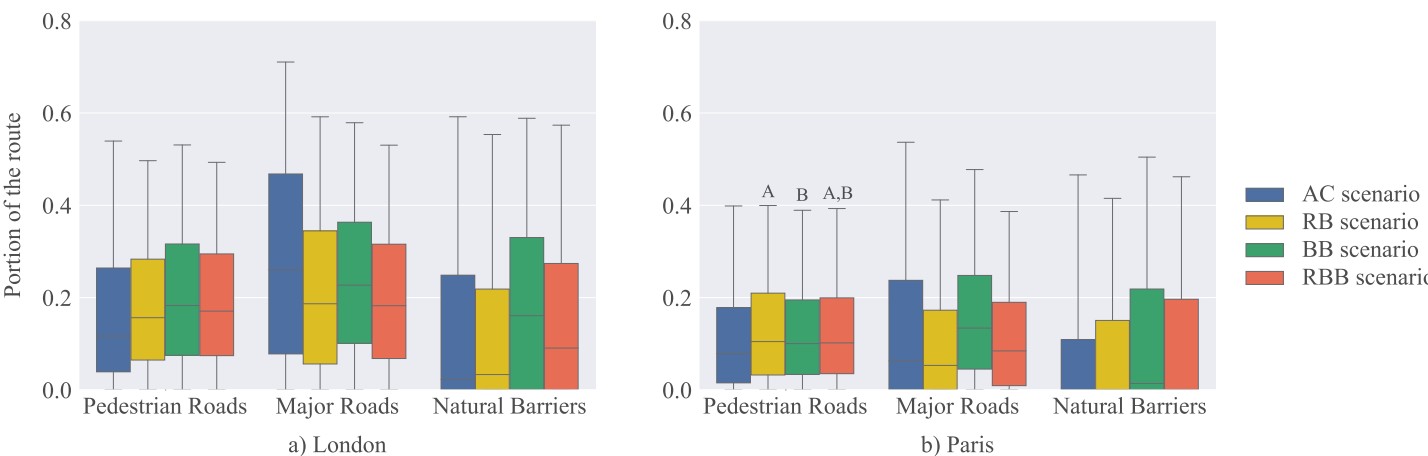

**Fig 9. Portion of the routes of the ABM scenarios walked along pedestrian roads, major roads and natural barriers.** Boxes that share the same letter indicate not statistically significant differences (Games-Howell post-hoc test). *AC*: Least cumulative angular change scenario; *RB*: Region-based scenario; *BB*: Barrier-based scenario; *RBB*: Region-barrier based scenario.

and parks played a role as in the BB scenario. Even though regionalisation processes might have induced the agents to cross the rivers rather than walking along them (RB scenario), the introduction of barriers directed the agents to take intra-regional detours aside rivers or within parks. The south bank of the Thames, in London, was walked by up 3.1% of the agents, the other side by 3.0% of the agents. Whereas the agent volumes aside the south-west of the Thames are alike the ones resulting from the AC and RB scenarios, streets along the east river-banks, the canal (north-east), and in proximity of the parks inherited high volumes from the BB scenario. Similar observations emerge from Paris, where the riverbanks come to light in their entire extent and the primary road in the south-east is noticeable in its continuity, as in the BB scenario.

Agents in the RBB scenario widely distributed across the street network, even more than the RB scenario. The region-barrier based route choice model proves to be the most "equal" routing mechanism in the two cities: the specific combination of regions and barriers further reduces the Gini coefficient arising from the RB scenario (London: -0.12; Paris: -0.08, from the AC scenario). Moreover, in London, the RBB scenario is characterised by the lowest portion walked along major roads (18%, Fig 9) and the second highest portion of pedestrian roads over the routes (17%), after the BB scenario (18%). In Paris, no significant differences as regards the share of pedestrian roads between the RB-RBB, and the BB-RBB scenarios were found. Besides, in the RBB scenario, the portions walked along natural barriers are generally low (median 0%, mean 12%), as already in the BB scenario (median 1%, mean 12%).

Overall, Paris is featured by a lower percentage of major roads (6% of the entire street network) as compared to London (12% of the network). In London, in the RB, BB and RBB scenarios, agents recur to such roads, attractive by design when minimising cumulative angular change, in lower portions as compared to the AC scenario. In Paris, however, major roads constitute a lower portion of the routes; only the BB scenario stands out in this regard. The two cities have a similar percentage of pedestrian roads (25% London, 24% Paris), yet all the route choice models appear to be more sensitive to pedestrian roads in London than in Paris. Considering that the sidewalk design nor a preference for certain road types were modelled, it is remarkable that both in London and Paris all the new route choice models induced agents to navigate more often along pedestrian roads as compared to the least cumulative angular change approach. Apart from the BB scenario, which resulted to be far from realistically representing pedestrian trips (see section 4.1), the RBB scenario's agents walked more often along pedestrian roads (London), within parks or next to water bodies (London and Paris). Notwithstanding, the region-based route choice model already generates routes featured by pedestrian roads almost (London) or as much (Paris) as the RBB scenario.

## 4.3 Limitations and future work

The main limitations of this work pertain a) the agents' spatial knowledge, b) the definition of regions and c) the absence of a calibration and validation of the ABM with observational data, as explained in the following. Agents were assumed to have a fixed knowledge about the environment. Although relationships between segments were represented imprecisely, by including error models, the enrichment of spatial knowledge with experience was not modelled. Furthermore, knowledge about the region was deterministic. We aim at accounting for differences in spatial knowledge in future work: agents may be endowed with differentiated symbolic representations of the city that account for the fact that only certain urban elements are known [116], on account of previous knowledge or differentiated spatial skills. In this sense, agents may be distinguished by goals, individual characteristics, and the likelihood to resort to certain route choice strategies rather than others.

Additionally, the regional subdivision employed in this work was static; it did not differ between agents on the basis of their characteristics (e.g. goals, demographic information, etc.), the trip, nor over time. An agent would always use the same regional structure of the city when formulating a path between two locations. In future work, a multi-level partitioning of the urban environment can be performed prior to the simulation so as to devise a hierarchically-nested subdivision, with different levels of granularity. In such a way, a more granular subdivision may be preferred by the agent, depending on the route characteristics. A similar approach has been devised with nested Voronoi partitions, to describe landmarks' *field of dominance*, in route descriptions generation for GI services [117]. The perception of regions varies as the explorer's frame of reference changes, depending on the length of the trip and other environmental factors.

Finally, the collection of micro-level observational data could support calibrating the interplay of different urban elements (e.g. regions, barriers, landmarks), for different types of agents, on the basis, for instance, of differences in spatial knowledge and spatial context. At the same time, by using macro-level data regarding the distribution of pedestrians across the street network, one can evaluate the performance of different route choice models and assess to what extent they can capture pedestrian movement flows. Along these lines, the modeller can establish more rigorously whether further refinements in the agents' cognitive architecture deepens the knowledge about pedestrian behaviour. WiFi hot-spots [118] or manual counts [36] have been used to obtain pedestrian volumes across urban areas and validate simulation models of pedestrian behaviour. We deem that such process is especially crucial when prediction, rather than the understanding of a certain phenomenon, as in our case, is the main purpose of the model. When studying the dynamics of complex systems, exploratory models can provide valuable insights by showing how divergent agents' behavioural architectures result in different emergent macro-level patterns [116, 119].

## 5 Conclusion

This paper explored and discussed the distribution of pedestrian movement flows across an urban environment, resulting from the incorporation of perceived urban subdivisions in pedestrian simulation. In light of research on the interaction between urban form and cognitive mapping [3, 50], the hierarchical nature of spatial knowledge [21, 39], evidence stemming from neuroscience on the role of barriers in spatial cognition [56, 57] and the fragmentation of cognitive representations of space [58, 59], we claim that information about regions and barriers should be included in route choice models for pedestrian simulations. We developed an ABM of pedestrian movement in urban spaces and devised four different scenarios to incrementally assess the effect of adding region and barrier information to their corresponding route choice models: a) currently prevailing least cumulative angular change minimisation, b) regions, c) barriers, and d) regions and barriers. The region-based approach entails two planning levels where the subdivision of the urban environment is used to depict a coarse initial plan that guides more granular intra-regional decisions [22, 32]. The barrier-based approach contemplates 'course-adjustments' [120] by encouraging navigation along and through rivers and parks, whilst avoiding walking along major roads and railways.

Based on two case studies, London and Paris, we sought to answer the following research questions: a) How plausible are the routes formulated by means of the different route choice models? b) How does the incorporation of regions and barriers affect the dispersion of pedestrians over the city? The deviation ratio from the corresponding road distance shortest path in the RBB scenario was found to be consistent with the findings reported in [97] and further considerations on the influence of environmental preferences in pedestrians [98, 110, 108].

The barrier-based scenario displayed an excessively large deviation, whereas the cost-based scenario showed to be somewhat implausible as featured by a low deviation ratio from the shortest path. The number of traversed regions and the length of the routes proved to partly explain the variation in the deviation ratio values in almost all the scenarios, both in Paris and in London.

At the street network level, the cumulative angular change scenario gave rise to highly clustered patterns. In London, agents distributed mostly across few major roads, not using many pedestrian roads. Similar patterns emerged in Paris, to a lesser extent. The French capital demonstrate to have a morphological structure that does not polarise agents into just few roads, even when agents minimise cumulative angular change. The introduction of regions brought about a much more dispersed pattern, both in London and Paris. Agents resorted to detours and walked through junctions whose salience derives from their status of transition nodes between regions. Furthermore, the presence of barriers yielded high volumes of agents along water bodies and green areas, which resulted in a higher usage of pedestrian roads, especially in London.

In view of these results, the region and the region-barrier based scenarios generated what we deem are more plausible pedestrian movement flows than the ones emerging from a prevailing cost-based route choice model. On the one hand, the incorporation of a region-based coarse plan in the route planning process produces more realistic distributions across the city and its different areas; on the other hand, barriers provoke concentrations of agents along pedestrian roads, nearby attractive natural areas and, at times, at major roads' junctions, generating routes that realistically deviate from the shortest path.

While plenty of ABMs for pedestrian movement in indoor contexts have been advanced [see: 121–123], models dealing with large number of people in urban contexts are rarer [35, 36, 82]. In this context, our model is specifically intended to understand pedestrian dynamics at the urban scale. It provides insights about locations featured by a high concentration of pedestrians, thus allowing urban planners and public organisations to consider interventions at the street level (e.g. sidewalk width and pedestrian facilities) and with regard to the urban configuration (e.g. severance caused by barriers in certain areas, absence of pedestrian paths, configurational complexity). As such, the model facilitates the tuning of different parameters so as to accommodate different research purposes, geographical characteristics and type of agents. Lastly, although an evaluation with macro-level observational data could enhance the understanding of pedestrian dynamics, this work may guide further research in route choice behaviour and shed light on the interplay between different urban elements in shaping humans' cognitive maps and navigation processes.

## Supporting information

**S1 Appendix.**
(PDF)

## Author Contributions

**Conceptualization:** Gabriele Filomena.

**Formal analysis:** Gabriele Filomena, Judith A. Verstegen.

**Investigation:** Gabriele Filomena.

**Methodology:** Gabriele Filomena, Ed Manley.

**Software:** Gabriele Filomena.

**Supervision:** Ed Manley, Judith A. Verstegen.

**Visualization:** Gabriele Filomena.

**Writing – original draft:** Gabriele Filomena.

**Writing – review & editing:** Gabriele Filomena, Ed Manley, Judith A. Verstegen.

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
