## [Decision Letter · Decision Letter 0]

4 Sep 2020

PONE-D-20-21441

Perception of urban subdivisions in pedestrian movement simulation

PLOS ONE

Dear Dr. Filomena,

Thank you for submitting your manuscript to PLOS ONE. After careful consideration, we feel that it has merit but does not fully meet PLOS ONE’s publication criteria as it currently stands. Therefore, we invite you to submit a revised version of the manuscript that addresses the points raised during the review process.

We look forward to receiving your revised manuscript.

Kind regards,

Nan Zheng

Academic Editor

PLOS ONE

Journal Requirements:

3. We note that Figures 2, 3, 4 and S1Fig1 in your submission contain map images which may be copyrighted. All PLOS content is published under the Creative Commons Attribution License (CC BY 4.0), which means that the manuscript, images, and Supporting Information files will be freely available online, and any third party is permitted to access, download, copy, distribute, and use these materials in any way, even commercially, with proper attribution. For these reasons, we cannot publish previously copyrighted maps or satellite images created using proprietary data, such as Google software (Google Maps, Street View, and Earth). For more information, see our copyright guidelines: http://journals.plos.org/plosone/s/licenses-and-copyright.

3.1.    You may seek permission from the original copyright holder of Figures 2, 3, 4 and S1Fig1 to publish the content specifically under the CC BY 4.0 license. 

3.2.    If you are unable to obtain permission from the original copyright holder to publish these figures under the CC BY 4.0 license or if the copyright holder’s requirements are incompatible with the CC BY 4.0 license, please either i) remove the figure or ii) supply a replacement figure that complies with the CC BY 4.0 license. Please check copyright information on all replacement figures and update the figure caption with source information. If applicable, please specify in the figure caption text when a figure is similar but not identical to the original image and is therefore for illustrative purposes only.

Reviewers' comments:

Reviewer's Responses to Questions

**Comments to the Author**

1. Is the manuscript technically sound, and do the data support the conclusions?

Reviewer #1: Partly

Reviewer #2: Partly

Reviewer #3: Partly

2. Has the statistical analysis been performed appropriately and rigorously? 

Reviewer #1: Yes

Reviewer #2: No

Reviewer #3: Yes

3. Have the authors made all data underlying the findings in their manuscript fully available?

Reviewer #1: Yes

Reviewer #2: Yes

Reviewer #3: Yes

4. Is the manuscript presented in an intelligible fashion and written in standard English?

Reviewer #1: Yes

Reviewer #2: Yes

Reviewer #3: No

5. Review Comments to the Author

Reviewer #1: This paper presents an agent based model that can be incorporated in pedestrian simulation models (mainly for route choice). The original feature of the model, as authors argue, is that it considers the cognitive processes obtained from the perception of regions and barriers. The paper is well written, concepts and results are nicely presented.

Pedestrian simulation tools (strategic, tactical, and operational) can be used in many purposes, e.g., to study evacuations, navigation, pedestrian behavior, etc. Some discussions are needed to explain the applications of the modeling framework presented in this study. Because depending on the application the models as well as the parameters may be different.

This paper only presents the outcomes of the simulations. This is one serious limitation of this work, however, authors have discussed such points in Section 4.3.

Route lengths presented in Table 1 seems a bit longer for “walking” scenarios and these lengths should be verified with real world data. Or some further discussions are needed to explain under which circumstances people walk such long distances.

Route choice is rather probabilistic. But, I cannot see any such point in this paper. Further, what is the sensitivity of the modelling framework? I.e., how the choice probability changes depending on the personal characteristics and changes in perception of the ‘barriers’?

Reviewer #2: 1. For normal distribution, variance is usually denoted as \\sigma^2

2. I would like authors to write (1)—(3) in the similar fashion. E.g., either all using \\sigma_f or all using \\sigma^2

3. I can understand the mean of Z_f shall be smaller than 1 in (2). Why do author choose 0.70? Is it a convention?

Reviewer #3: The paper offers a model to estimate pedestrian movement with region-barrier based route choice model. The motion is essentially influenced by regions and barriers. The model uses agent-based modeling (ABM) approach. The paper is not acceptable in this version, on my opinion. Major comments follow.

1.Firstly, the literature review is not clearly offered. It seems to be divided in Section 2, and 3 (eg.line 233) but the reader cannot have a complete overview of the problems. I suggest to: 1) use the introduction to concisely point out the problem and the proposed paper solution; 2) merge all the literature review parts in a unique section by dividing the main fields and clearly providing the limits of the current studies; 3) provide a clear connection between each statement and the related references. I also suggest the authors to look existing works agent-based modeling for user movement.

2. Abstract: (Line 3-5) Are you sure that there is no existing ABM for pedestrian movement considering the cognitive processes of human ? I strongly suggest the authors to check literature again.

3. Section 3: Please motivate why ABM is ideal for modeling the pedestrian movement. What is the relation between cognitive issues and agents ?

4. Section 3: (Line 262-263) Please give (if possible) specific references and motivation to 70 degree angle. How did you determine ?

5. Section 3: Fig.3- What is the identified sub-goal (green point) ? Can you explain ?

6. Figures that use black background, makes difficult to understand the origin and destination points. They are really eye-straining, please consider to revise.

But mainly, I am not really sure that all the research involves cognitive processes of human to the movement. There is no qualitative research like observation or interview etc. Can you explain more about how you integrate cognitive maps of pedestrian to the simulation model ?

Also there is a major problem about the validation of the simulation model. The model has to be validated with approved approaches.

6. PLOS authors have the option to publish the peer review history of their article (what does this mean?). If published, this will include your full peer review and any attached files.

Reviewer #1: No

Reviewer #2: No

Reviewer #3: No

---

## [Author Response · Author response to Decision Letter 0]

2 Nov 2020

Dear editor,

Thank you for considering our paper for publication. We have addressed the comments of the reviewer and changed the manuscript accordingly. Please find below our responses and a detailed description of the related changes in the manuscript per comment of the reviewer. Line numbers mentioned in the responses refer to the line numbers in the revised manuscript. In the revised manuscript (“Revised manuscript with track changes”), changes are highlighted in blue.

In relation to the copyright remarks on Figure 2, 3, 4 and S1Fig1, these have been prepared by the authors by using OpenStreetMapData. They are not reproduction of existing figures. The data’s copyright holder states that “OpenStreetMap® is open data, licensed under the Open Data Commons Open Database License (ODbL) by the OpenStreetMap Foundation (OSMF). You are free to copy, distribute, transmit and adapt our data, as long as you credit OpenStreetMap and its contributors. If you alter or build upon our data, you may distribute the result only under the same licence.” 

The rendering was completely performed by the author, after having processed the OpenStreetMap geodata (simplification of the geometries). No OpenStreetMap tiles nor maps have been used. The caption of these figures now reports explicitly the data source.

We are thankful for the constructive feedback from you and the reviewers and we believe that these have improved our manuscript considerably. We hope you will reconsider our manuscript for publication.

Yours sincerely,

Gabriele Filomena, on behalf of the authors

Reviewer #1:

Reviewer’s comment: This paper presents an agent based model that can be incorporated in pedestrian simulation models (mainly for route choice). The original feature of the model, as authors argue, is that it considers the cognitive processes obtained from the perception of regions and barriers. The paper is well written, concepts and results are nicely presented. Pedestrian simulation tools (strategic, tactical, and operational) can be used in many purposes, e.g., to study evacuations, navigation, pedestrian behavior, etc. Some discussions are needed to explain the applications of the modeling framework presented in this study. Because depending on the application the models as well as the parameters may be different.

Author’s response: Thank you for your kind words and comments. We agree that the potential applications of our proposed framework required clarification. We have discussed more extensively the purposes of our work, within other possible applications of pedestrian simulation models (see the conclusion, lines: 773-776, 781-783).

Reviewer’s comment: This paper only presents the outcomes of the simulations. This is one serious limitation of this work, however, authors have discussed such points in Section 4.3. Route lengths presented in Table 1 seems a bit longer for “walking” scenarios and these lengths should be verified with real world data. Or some further discussions are needed to explain under which circumstances people walk such long distances.

Author’s response: Thank you for your suggestion. We have selected origin-destination pairs that cover the entire study region, and ensure agents have opportunities to make errors. Nevertheless, we show that deviation from the road distance shortest-path is consistent with empirical evidences provided in other studies (section 4.1). We do agree however the necessity to indicate what kind of individuals are modelled. We extended section 3.1 (lines: 231-237). Furthermore, we have expanded the discussion on the importance of evaluating models with real world data in the limitations and future work section of the discussion (lines: 719-724).

Reviewer’s comment: Route choice is rather probabilistic. But, I cannot see any such point in this paper. Further, what is the sensitivity of the modelling framework? I.e., how the choice probability changes depending on the personal characteristics and changes in perception of the ‘barriers’?

Author’s response: Thanks for this remark. Some probabilistic aspects are already contemplated in our model at the agent level. When two routes have similar actual costs, an agent will not always pick the shortest one, because we have stochastic variables for the perceived cost of the road segments (see equations 1, 2, 3). We have added a sentence below the equations to better explain this asset to the reader (lines 270-272). At a higher level, we agree that certain groups of individuals (e.g. expert pedestrians, commuters, etc.) may be more likely to resort to certain route-choice models rather than others, conversely to other groups. We plan to further explore such differences in successive work and we have reported this limitation in the discussion (see: section 4.3, lines: 692-694).

Reviewer #2: 

Reviewer’s comment:

1. For normal distribution, variance is usually denoted as \\sigma^2

2. I would like authors to write (1)—(3) in the similar fashion. E.g., either all using \\sigma_f or all using \\sigma^2

Author’s response: Thanks, we have changed the notation of the equations by employing sigma^2 and a homogeneous notation, as suggested.

Reviewer’s comment: 3. I can understand the mean of Z_f shall be smaller than 1 in (2). Why do author choose 0.70? Is it a convention?

Author’s response: We have chosen 0.70 in (2) and 1.30 (3) so to represent underestimation and overestimation of distance along positive and negative barriers, respectively; in other words, to manipulate their attractiveness. A sensitivity analysis could be conducted with other combination of values (e.g. 0.60 and 1.40, 0.80 and 1.20, etc.) so to further assess the influence of this parameter. We subjectively chose these values as they proved to elicit differences in the agents’ behaviour without affecting the plausibility of the routes (we have now reported this aspect in lines: 376-378).

Reviewer #3: 

Reviewer’s comment: The paper offers a model to estimate pedestrian movement with region-barrier based route choice model. The motion is essentially influenced by regions and barriers. The model uses agent-based modeling (ABM) approach. The paper is not acceptable in this version, on my opinion. Major comments follow.

Author’s response: Thank you for offering several points of reflection. We hope you will find the paper stronger, given our changes.

Reviewer’s comment: 1.Firstly, the literature review is not clearly offered. It seems to be divided in Section 2, and 3 (eg. line 233) but the reader cannot have a complete overview of the problems. I suggest to: 1) use the introduction to concisely point out the problem and the proposed paper solution; 2) merge all the literature review parts in a unique section by dividing the main fields and clearly providing the limits of the current studies; 3) provide a clear connection between each statement and the related references. I also suggest the authors to look existing works agent-based modeling for user movement.

Author’s response: We have reshaped the literature review to make it more cohesive (main changes, lines: 63-82, 103-136, 139-155, 179-188). In Section 2, the first 2 subsections motivate the relevance of regions and barriers in spatial cognition (2.1 – in cognitive mapping, 2.2. – in route choice behaviour) and therefore why we deem important including them in ABM for pedestrian simulation. We have added a new subsection (2.3, the original text has been integrated in 2.1 and 2.2 or removed if superfluous) that reports about existing ABMs for pedestrian simulation in urban environments and shows what gaps emerge, in relation to research reported in 2.1 and 2.2. 

References in the methodology section (section 3) have been provided to make the links between the methodology and the background explicit; when necessary, we have readjusted such cross-references so that the link is clearer.

Reviewer’s comment: 2. Abstract: (Line 3-5) Are you sure that there is no existing ABM for pedestrian movement considering the cognitive processes of humans? I strongly suggest the authors to check literature again.

Author’s response: In the sentence the reviewer refers to, ‘These cognitive mapping processes’ refers to ‘regionalisation processes and the identification of separating elements’ in the previous sentence (line 1 and 2, abstract). Although other cognitive processes may have been modelled in existing approaches for pedestrian simulation (see section 2.3, now devoted to existing ABM approaches), regionalisation processes and the identification of barriers haven’t, at least to our knowledge. We have amended the sentence to make clearer that we refer to such specific cognitive mapping processes instead of cognitive mapping processes in general.

Reviewer’s comment: 3. Section 3: Please motivate why ABM is ideal for modeling the pedestrian movement. What is the relation between cognitive issues and agents?

Author’s response: Thank you for the suggestion. We have further elaborated on this in section 2.3 (lines 191-206). 

Reviewer’s comment: 4. Section 3: (Line 262-263) Please give (if possible) specific references and motivation to 70 degree angle. How did you determine?

Author’s response: This was determined on the basis of an existing simulation models for taxi drivers. We have adapted this constraint for pedestrian navigation to reflect strongly goal directed behaviour. Further explanation is now provided in the text (lines 313-318).

Reviewer’s comment: 5. Section 3: Fig.3- What is the identified sub-goal (green point)? Can you explain?

Author’s response: This is a barrier sub-goal, namely the node that is the closest to the current location of the agent that is also aside the previously identified barrier (definition in line 350). We have added the word ‘barrier’ to the caption so to make it easier to refer to the definition.

Reviewer’s comment: 6. Figures that use black background, makes difficult to understand the origin and destination points. They are really eye-straining, please consider to revise.

Author’s response: We were surprised by the comment that the black background is eye-straining. In fact, programmers nowadays use a black background for their code editors in order to reduce eye-strain; the opposite of the reviewer’s experience. We had tested other solutions before submission (including white background figures), but we deem the one currently embraced to be the most effective for showing differences across scenarios and the two cities. For the figures in section 3, we believe the black background allows to better detect differences amongst the elements represented. 

Reviewer’s comment: But mainly, I am not really sure that all the research involves cognitive processes of human to the movement. There is no qualitative research like observation or interview etc. Also there is a major problem about the validation of the simulation model. The model has to be validated with approved approaches.

Author’s response: We appreciate your concern. Our work attempts to computationally model cognitive processes that involve the usage of regions and barriers in route-choice behaviour and in the perception of the urban environment. Of course, by the definition of the word model, incorporating such processes in a model entails a simplification of human spatial cognition and its nuances. We have employed existing observations from literature to evaluate the plausibility of the routes generated by the model (section 2.3). Such studies have been conducted with large samples of pedestrians. We agree that a wider qualitative validation of the assumptions on which the model is based – e.g. in relation to individual differences, preferences for certain elements, etc. - would make the approach stronger. This is planned for future work. 

However, the scope of this paper pertains to the understanding of pedestrian dynamics and route-choice behaviour rather than a precision prediction of pedestrians’ distributions. For the latter, a rigorous validation would be, we agree, very much necessary. Having said that, ABM is still characterised by a large debate around validation and verification methods. Micro and macro approaches have not been standardized and, as such, there are no approved approaches to evaluate ABM (see for example Thober, et. all (2017) 'Agent-Based Modelling of Social-Ecological Systems: Achievements, Challenges, and a Way Forward' Journal of Artificial Societies and Social Simulation 20 (2) 8). Moreover, exploratory-modelling is of major importance to demonstrate the effect of different modelling approaches, especially in the absence of representative data. We have expanded the discussion on this issue in the limitations and future work section of the discussion (lines: 416-418, 719-724).

---

## [Decision Letter · Decision Letter 1]

3 Dec 2020

Perception of urban subdivisions in pedestrian movement simulation

PONE-D-20-21441R1

Dear Dr. Filomena,

We’re pleased to inform you that your manuscript has been judged scientifically suitable for publication and will be formally accepted for publication once it meets all outstanding technical requirements.

Kind regards,

Nan Zheng

Academic Editor

PLOS ONE

Additional Editor Comments (optional):

Reviewers' comments:

Reviewer's Responses to Questions

**Comments to the Author**

1. If the authors have adequately addressed your comments raised in a previous round of review and you feel that this manuscript is now acceptable for publication, you may indicate that here to bypass the “Comments to the Author” section, enter your conflict of interest statement in the “Confidential to Editor” section, and submit your "Accept" recommendation.

Reviewer #1: All comments have been addressed

Reviewer #2: All comments have been addressed

2. Is the manuscript technically sound, and do the data support the conclusions?

Reviewer #1: Yes

Reviewer #2: Yes

3. Has the statistical analysis been performed appropriately and rigorously? 

Reviewer #1: N/A

Reviewer #2: Yes

4. Have the authors made all data underlying the findings in their manuscript fully available?

Reviewer #1: Yes

Reviewer #2: Yes

5. Is the manuscript presented in an intelligible fashion and written in standard English?

Reviewer #1: Yes

Reviewer #2: Yes

6. Review Comments to the Author

Reviewer #1: Authors have improved and updated the paper considering reviewers' comments. Further, they have responded the queries made by reviewers.

I have no further comments.

Reviewer #2: My questions are all answered. I do not have more question. I am typing to reaching the minimum character count.

7. PLOS authors have the option to publish the peer review history of their article (what does this mean?). If published, this will include your full peer review and any attached files.

Reviewer #1: No

Reviewer #2: No

---

## [Editor Report · Acceptance letter]

14 Dec 2020

PONE-D-20-21441R1 

Perception of urban subdivisions in pedestrian movement simulation 

Dear Dr. Filomena:

I'm pleased to inform you that your manuscript has been deemed suitable for publication in PLOS ONE. Congratulations! Your manuscript is now with our production department. 

Kind regards, 

on behalf of

Dr. Nan Zheng 

Academic Editor

PLOS ONE